# The Impacts of Combined Blood Flow Restriction Training and Betaine Supplementation on One-Leg Press Muscular Endurance, Exercise-Associated Lactate Concentrations, Serum Metabolic Biomarkers, and Hypoxia-Inducible Factor-1α Gene Expression

**DOI:** 10.3390/nu14235040

**Published:** 2022-11-27

**Authors:** Steven B. Machek, Dillon R. Harris, Emilia E. Zawieja, Jeffery L. Heileson, Dylan T. Wilburn, Anna Radziejewska, Agata Chmurzynska, Jason M. Cholewa, Darryn S. Willoughby

**Affiliations:** 1Department of Health, Human Performance, and Recreation, Robbins College of Health and Human Sciences, Baylor University, Waco, TX 76706, USA; 2Kinesiology Department, College of Health Sciences and Human Services, California State University, Monterey Bay, Seaside, CA 93955, USA; 3Department of Human Nutrition and Dietetics, The Poznań University of Life Sciences, 60-637 Poznań, Poland; 4Nutrition Services Division, Walter Reed National Military Medical Center, Bethesda, MD 20814, USA; 5Exercise Physiology Department, University of Lynchburg, Lynchburg, VA 24501, USA; 6School of Exercise and Sport Science, Mayborn College of Health Sciences, University of Mary Hardin-Baylor, Belton, TX 76513, USA

**Keywords:** arterial occlusion pressure (AOP), blood flow restriction training (BFR), betaine supplementation, resistance training, growth hormone (GH), homocysteine, Hypoxia Inducible Factor-1α (HIF-1α), insulin-like growth factor-1 (IGF-1)

## Abstract

The purpose of this investigation was to compare the impacts of a potential blood flow restriction (BFR)-betaine synergy on one-leg press performance, lactate concentrations, and exercise-associated biomarkers. Eighteen recreationally trained males (25 ± 5 y) were randomized to supplement 6 g/day of either betaine anhydrous (BET) or cellulose placebo (PLA) for 14 days. Subsequently, subjects performed four standardized sets of one-leg press and two additional sets to muscular failure on both legs (BFR [LL-BFR; 20% 1RM at 80% arterial occlusion pressure] and high-load [HL; 70% 1RM]). Toe-tip lactate concentrations were sampled before (PRE), as well as immediately (POST0), 30 min (POST30M), and 3 h (POST3H) post-exercise. Serum homocysteine (HCY), growth hormone (GH) and insulin-like growth factor-1 concentrations were additionally assessed at PRE and POST30M. Analysis failed to detect any significant between-supplement differences for total repetitions completed. Baseline lactate changes (∆) were significantly elevated from POST0 to POST30 and from POST30 to POST3H (*p* < 0.05), whereby HL additionally demonstrated significantly higher ∆Lactate versus LL-BFR (*p* < 0.001) at POST3H. Although serum ∆GH was not significantly impacted by supplement or condition, serum ∆IGF-1 was significantly (*p* = 0.042) higher in BET versus PLA and serum ∆HCY was greater in HL relative to LL-BFR (*p* = 0.044). Although these data fail to support a BFR-betaine synergy, they otherwise support betaine’s anabolic potential.

## 1. Introduction

Skeletal muscle is an incredibly adaptable tissue, responding to various stimuli that subsequently enhance functions in power, strength, and endurance [1,2]. Two such stimuli touted as ergogenic aids are blood flow restriction (BFR) training and betaine (N-N-N-trimethylglycine) supplementation. Introduced in the 1960′s under the Kaatsu moniker, BFR employs partial-arterial and full-venous occlusion of the targeted limb(s) to elicit hypertrophic and commensurate strength benefits, as well as beneficial vascular adaptations [2,3,4]. Uniquely, BFR research has produced these various positive outcomes using low-intensity exercise such as walking and low-load (~20–50% one repetition maximum [1RM]) resistance exercise [5]. This is especially important amidst demographics that are contraindicated to conventional un-cuffed, higher-load training, whereby the American College of Sports Medicine (ACSM) recommends lifting approximately 70%1RM to achieve hypertrophy [2,6]. BFR mechanistically operates via producing a localized, hypoxic, and subsequently acidotic cellular environment, concomitantly shifting towards anaerobic metabolism [3,5]. This acute metabolic alteration mediates the preferential activation of high-threshold motor units and associated fast-twitch muscle fibers that would otherwise remain largely underutilized in a low-intensity/load exercise scenario [5,7]. Furthermore, greater reliance on glycolysis-derived adenosine triphosphate (ATP) facilitates the lactate dehydrogenase A (LDHA)-dependent reduction of pyruvate to lactate. This reaction simultaneously combats ATP hydrolysis-mediated acidosis and permits glycolysis to continue energy provision via nicotinamide adenine dinucleotide (NAD+) restoration [3,8,9]. BFR-mediated hypoxia and concomitant changes in pH, potassium, and hydrogen ions, are ostensibly claimed to both stimulate the secretion of growth hormone (GH) and augment hypoxia-inducible factor (HIF) DNA binding [2]. Lactate specifically has been additionally credited to enhance serum growth hormone both exponentially and sustainably (60–90 min post exercise) with exercise crossing lactate threshold, whereby the latter is thought to mediate hypertrophy via associated insulin-like growth factor (IGF-1) release [2,10]. Nevertheless, the preceding literature on this potential hypoxia-mediated phenomenon is wholly unclear, whereby there is mixed data supporting BFR-specific increases in lactate, GH, and IGF-1 [11,12,13,14,15,16]. Furthermore, HIF is a heterotrimeric transcription factor comprised of HIF-1α and the constitutively expressed HIF-1ß. HIF-1α is normally targeted by the von Hippel-Lindau E3 ubiquitin ligase and degraded by the 26S proteasome following hydroxylation of proline (Pro^402^ and Pro^564^) residues under normoxic scenarios [3]. Conversely, HIF-1α is able to translocate across the nuclear membrane under hypoxic conditions and dimerize with constitutively expressed HIF-1ß to form an active transcription factor; this active DNA binding protein then binds to hypoxia response elements in the promoter region upstream of hypoxia-related genes to ultimately influence the transcription of several associated genes, including those involved in lactate metabolism and aerobic capacity [3]. While the aforementioned mechanisms and data otherwise substantiate BFR as a suitable alternative to conventional high-load training, perhaps minute augmentations to this training practice could enhance it beyond its predecessor.

Betaine is a choline-derived compound found in foods such as sugar beets and spinach, but has recently come into the sports nutrition spotlight [17,18]. Specifically, it has been implemented in ranging doses (2.5–6.0 g/day) to largely augment muscular endurance, as well as enhance skeletal muscle accretion with concomitantly reduced adiposity [18,19,20,21]. Betaine mainly functions to complement the roles of folate and vitamin B12, facilitating the transmethylation of homocysteine (HCY) to methionine via enzymatic action of betaine-homocysteine methyltransferase (BMHT) [22]. Prevention of high HCY concentrations (hyperhomocysteinemia) is imperative for decreasing atherosclerosis risk, oxidative stress, and protein damage [17,21,23]. Incidentally, betaine also serves an important role as an organic osmolyte, relieving osmotic stress in largely hypertonic environments or tissues with a high methyl metabolism such as the kidney and liver, respectively [24,25,26]. This compound’s macromolecule structure also permits it to exist “compatibly” in relatively high concentrations without modifying intracellular protein functions [25,27]. As an intracellular osmolyte, betaine can stabilize protein structures by enhancing folded aqueous protein hydrogen bonds to ultimately protect them against denaturation [26,27,28]. It is therefore mechanistically plausible—given the aforementioned beneficial betaine-specific benefits—that supplementation may support the mechanistic action of BFR training.

Both BFR exercise and betaine supplementation are touted for their ability to distinctly augment both intracellular and whole skeletal muscle level hypertrophic outcomes [2]. Moreover, a combined BFR-betaine supplementation synergy may uniquely support the metabolic environment and mechanical tension that are widely accepted to mediate BFR’s skeletal muscle hypertrophy mechanisms [7,29]. The latter commonly employs low-loads and brief (30–60 s) rest periods, along with standardized set and repetition schemes in lieu of multiple sets to failure; this ultimately promotes an adequate mechanical stimulus for subsequent adaptation [30]. Therefore, the benefits of additional betaine supplementation are two-fold: skeletal muscle saturated with osmolytic betaine may protect muscle intracellular proteins against BFR exercise-associated acidotic denaturation, ultimately potentiating greater mechanical tension via attenuated metabolic fatigue and concomitantly prolonged contraction [26,27,28]. Betaine supplementation is also reported to increase red blood cell count, hemoglobin count, and subsequent hematocrit percentage, potentially due to its role in facilitating the folate-dependent synthesis of purines and pyrimidines [22,26]. Consequently, perhaps the aforementioned enhancements in muscular performance can be compounded by betaine-specific oxygen carrying capacity augmentations. An increased workload facilitated by a BFR-betaine combination would hypothetically facilitate more repetitions and concomitantly facilitate increased glycolytic flux, subsequently generating additional lactate [3,8,9]. This lactate and the unique metabolic environment produced may then promote increased serum GH and commensurate IGF-1 [10,11]. Interestingly, betaine has been independently reported to increase GH and IGF-1 in humans and animal models [17]. Combined BFR and betaine supplementation therefore compound again, possibly providing further enhanced intracellular anabolic signaling [2]. The feasibility of this synergy and the potential economic advantage it may bestow to various demographics warrants investigation. Therefore, this study aimed to test the hypothesis that a combination of BFR resistance training and betaine supplementation would enhance repetitions to fatigue and facilitate an anabolic environment fostering growth. This would be further evidenced by relatively greater lactate concentrations, reduced serum HCY, as well as concomitantly elevated relative GH and IGF-1 levels compared to either isolated modality or control conditions. Lastly, we hypothesized that the potential BFR-betaine synergy and concomitantly augmented hypoxic local environment would facilitate enhanced relative *HIF-1A* gene expression compared to all other conditions.

## 2. Materials and Methods

### 2.1. Experimental Approach to the Problem

In this randomized, double-blind, placebo-controlled mixed-model (within-and-between subject design) investigation, participants visited the laboratory on three separate occasions in the following manner: visit 1 = entry/familiarization, medical/physical activity screening, and supplement/placebo pick-up, visit 2 = body composition assessment and determination of BFR cuff size and randomized-leg 1-repetition maximum (1RM) assessment of each leg employing the horizontal one-leg press exercise, and visit 3 = hematocrit/packed cell volume (PCV), arterial occlusion pressure (AOP) assessment, and resistance exercise, whereby each participant performed six sets of both (randomized and counter-balanced) high-intensity (HL; 70%1RM) and low-intensity BRF (LL-BFR; 20%1RM) exercise on the seated horizontal leg press. Visit 2 preceded visit 3 by 48-h, allowing participants adequate rest between 1RM testing prior and HL/LL-BFR resistance exercise. Both groups performed four sets of one-leg press exercise at 10 (HL) and 30, 15, 15, 15 (LL-BFR) repetitions; participants subsequently performed two additional sets to muscular failure. All rest intervals were strictly limited to 45-s. Blood samples were obtained from subjects before and 30 min after exercise, whereas skeletal muscle biopsy samples were collected before and 3 h after exercise. Lastly, toe-tip capillary lactate was assessed at all aforementioned time points, as well as immediately after each experimental exercise condition. A concise depiction of the study design and timeline are illustrated in Figure 1.

### 2.2. Participants

Eighteen apparently healthy, recreationally resistance-trained (modified from the American College of Sports Medicine recommendations; ≤30 min exercise, ≤3 days per week, over the last 3 months, as well as having a minimum of one day per week leg-focused exercise over the last year prior to the onset of the investigation) men between the ages of 18–35 volunteered to serve as subjects in this study [6]. Enrollment was open to men of all ethnicities. Women were not recruited due to sex-specific variations in methyl metabolism and tissue betaine concentration [18,31]. The use of blood thinning (e.g., Warfarin, Jantoven, etc.), heart, pulmonary, thyroid, antihypertensive, anti-hyperlipidemic, hypoglycemic, endocrinologic (e.g., prednisone, Ritalin, Adderall, etc.), or neuromuscular/neurological medications were further prohibited for eligibility. Furthermore, all subjects were required to have a resting (systolic and diastolic) blood pressure <140/90 mmHg and a resting heart rate <90 bpm following 10 min of seated rest in a temperature-controlled quiet room [32,33]. All eligible subjects signed university-approved informed consent documents and approval was granted by the Institutional Review Board for Human Subjects at Baylor University (reference #1676709, approval date: 1/26/2021). In addition, all experimental procedures involved in the study conformed to ethical considerations of the Helsinki Code.

### 2.3. Dietary Records

Subjects were required to record their dietary intake for 24 h prior to visit 1. All recalls were then evaluated by a board-certified registered dietitian (RD) with the Food Processor dietary assessment program (ESHA Research, Salem, OR, USA) to determine any relevant vitamin deficiencies. Considering specific b vitamin deficiencies may impair betaine skeletal muscle concentrations and general metabolism, a b vitamin complex (Nature Made Super B Energy Complex) was given to subjects when supplementation would mend any pertinent nutritional deficiencies [34]. Importantly, the prevalence of these deficiencies was evaluated from the aforementioned dietary records by our board-certified RD. Subjects administered the B vitamin complex were instructed to take one serving every day until their final visit. Otherwise, subjects were asked to not change their dietary habits during the study timeline. Subjects were additionally required to record their dietary macronutrient intake for 48 h prior to their final visit using the MyFitnessPal (San Francisco, CA, USA) mobile or desktop application and were instructed on how to use relevant features if unfamiliar with the modality. Dietary macronutrient (protein, carbohydrate, and fat), as well as fiber intake were recorded for all 48 h dietary records. All macronutrient and fiber data were subsequently averaged between days and normalized to weight (kg) for further statistical analysis.

### 2.4. Betaine Supplementation Protocol

In between the first and final visits, subjects continuously consumed 3 g/twice daily (6 g total per day; separated by ~12 h) either betaine anhydrous (BET; Vital Pharmaceuticals [VPX] inc., Weston, FL, USA) for 14 days to allow for skeletal muscle saturation or cellulose placebo (PLA; NutriCology, South Salt Lake, UT, USA) in matched doses and times [34,35,36]. Supplement preparation and distribution was done so in a double-blind and counterbalanced manner (9 subjects assigned to BET and 9 assigned to PLA), whereby both conditions were identically encapsulated (fine white powder in transparent gelatin capsules). Subjects returned their empty supplement containers on the last visit to visually confirm supplementation protocol adherence. As previously mentioned and similar to a previous investigation by our laboratory, subjects consumed a b-vitamin complex alongside their betaine/placebo supplementation if deemed necessary by the investigative team’s RD [34]. Subjects were asked to consume their last dose of either supplement 12 h from their last visit for standardization.

### 2.5. Body Composition Testing

Total body mass (kg) and height (cm) were determined on a standard dual beam balance scale (Detecto Bridgeview, IL, USA) during the screening visit. Upon arriving to the laboratory on the final visit in a fasted state (including caffeine intake), subject percent body fat, fat mass, and fat free mass were determined using dual-energy-x-ray-absorptiometry (DEXA) (Hologic Discovery Series W, Waltham, MA, USA). Quality control calibration procedures were performed on a spine phantom (Hologic X-CAIBER Model DPA/QDR-1 anthropometric spine phantom) and a density step calibration phantom prior to each testing session.

### 2.6. One-Repetition Maximum (1RM) Testing

To determine muscular strength and subsequent relative load prescriptions, subjects performed one-repetition maximum (1RM) tests on both right and left legs in randomized, crossover, and counterbalanced fashion on a horizontal one-leg press (Nebula Fitness Equipment, Scottsdale, AZ, USA) in accordance with NSCA recommendations on the second visit [37]. All subjects were required to have a minimum one-leg press—on both limbs—of at least 1× bodyweight, predicated on previously described strength standards and applying prior evidence on the leg press-specific bilateral deficit [38,39,40]. Furthermore, two days separated each subject’s second and third (final/experimental) visits. All subject exercise testing and protocols were completed using a standardized four-point tempo prescription that controls eccentric, amortization, concentric, and lift beginning (1-0-1-0) to standardize all repetitions. Additionally, horizontal one-leg press foot placement was recorded and held constant over all testing conditions to maintain consistency. To ensure subjects were moving through the full range of motion during each repetition, a goniometer was used to establish 90° of knee flexion on the leg press. Any excessive “bouncing” at the intersection of the eccentric and amortization phases was not counted as a successful repetition. Subjects warm-ups were standardized and adapted from the procedures used previously by Wallace et al. [41]. Briefly, subjects completed 10 repetitions at approximately 50% of their estimated 1RM; subsequently, subjects rested for 2 min before completing 5, 3, and 1 repetition(s) at approximately 70%, 80%, and 90% of their estimated 1RM, respectively. Load was then increased conservatively ~5–10%—as per lower-body exercise testing NSCA guidelines—and the subject attempted to lift the load for 1 repetition [37]. If the lift is successful, the subject rested for 3 min before attempting the next weight increment, and this procedure continued until the subject failed to complete the lift. The 1RM was recorded as the maximum weight that the subject is able to lift for a single repetition. All attempts 90% and above were blinded from the subject via weight coverings using similar methods to a prior investigation in our laboratory [42]. Lastly, subjects rested for 10 min prior to completing the crossover assessment on the contralateral limb.

### 2.7. Blood Flow Restriction Cuff Application and Arterial Occlusion Pressure (AOP)

A commercially available narrow-elastic (product size #4; length = 60.96 cm, width = 9.53 cm) BFR cuff (B3 Sciences, Frisco, TX, USA) was employed to elicit partial arterial occlusion for all LL-BFR conditions on the experimental/final visit. During the preceding visit, all subjects’ resting leg circumferences (specific to leg previously randomized to the LL-BFR condition and counter-balanced between subjects) were measured immediately distal to the inguinal crease and a distinct pen mark was made on the skin directly distal to the bottom portion of the cuff to ensure proper placement during their subsequent visit. Subject resting AOP was determined via a handheld Doppler (Sonotrax Vascular, Edan USA, San Diego, CA, USA) with a 8 MHz probe on the final/experimental. AOP assessment was performed immediately before the resistance exercise but prior to a 10 min supine resting period inside a quiet, temperature-controlled room; briefly, AOP was recorded as the point of increasing cuff pressure whereby the auditory ausultatory signals of the posterior tibial artery are no longer detectable [33,43]. Importantly, the handheld Doppler is validated against the gold standard Pulse Wave Doppler (r = 0.938, *R*^2^ = 0.879). As complete arterial occlusion may result in premature fatigue, local ischemia, and subsequently hindered performance, LL-BFR condition cuffs were standardized and inflated to 80% resting AOP as supported by previous literature [4]. Notably, AOP could not be determined for one subject, ostensibly due to his relatively smaller upper thigh circumference. Prioritizing standardized cuff pressure in lieu of homogenous cuff width, we employed a smaller cuff (product size #3; length = 50.80 cm; width = 7.62 cm) from the same aforementioned distributor and were subsequently able to assess AOP.

### 2.8. Resistance Exercise Protocol

During the final/experimental visit, subjects performed six sets of horizontal one-leg press in a condition-randomized, crossover, and between-subject counterbalanced manner. Furthermore, all subjects rested for 10 min prior to completing the crossover exercise condition on the contralateral limb. To minimize potential nutrient-mediated impacts on performance and/or assessed serum targets, subjects consumed a standardized nutrition bar (Power Bar^®^, Premier Nutrition Corporation, Kings Mountain, NC, USA [carbohydrate: 25 g, protein: 20 g, fat: 6 g, fiber 4 g]) 30 min prior to initiating the horizontal one-leg press warm-ups [44]. As one of our primary experimental variables, load varied between legs, whereby the subject’s starting leg and condition were randomized and between-subject counterbalanced. After a 10 min warm-up on a cycle ergometer, subjects additionally warmed-up based on the one-leg press protocol described by Clark et al. [45]. In brief, each subject lifted 50% of each leg’s previously determined 1RM for 10 repetition as additional modality-specific warm-up before continuing on to either HL or LL-BFR one-leg press exercise following a two-minute rest [45]. As previously described and performed in a previous investigation in our laboratory, subjects undergoing HL (70%1RM) performed 4 sets of 10 repetitions, whereas LL-BFR (20%1RM) had a repetition scheme of 30, 15, 15, 15 [46]. Both HL and LL-BFR conditions were subsequently followed by two sets to muscular failure and all sets were interspersed by 45 s of timed rest. The total number of repetitions across all sets were recorded and subsequently analyzed to determine any potential between-supplement differences. Similar to maximal testing, all experimental testing session loads were blinded to the subject prior to the warm-up and exercise protocol via weight coverings [42]. Immediately following the cessation of all six LL-BFR sets, cuff pressure was released and the device was removed.

### 2.9. Rated Perceived Exertion and Subjective Discomfort Scales

Rated perceived exertion (RPE) was assessed using a 6–20 scale, whereby “6” and “20” suggest no exertion and maximal exertion, respectively. Additionally, perceptual discomfort between conditions was assessed via the Borg CR10+ scale and baseline values were obtained immediately following the initial 10 min supine resting period prior to HL/LL-BFR warm-up exercise. The RPE and subjective discomfort scales were explained in detail akin to the methods described by Loenneke et al. [47] and Buckner et al. [33]. Briefly, the Borg CR10+ scale assesses subjective discomfort on a scale from 1 (no discomfort) to 10 (maximal discomfort); the latter rating is also anchored by the participant’s greatest memory of discomfort and therefore can be exceeded if the present methods exceed that experience [47]. Participants were consistently asked to describe their RPE and perceived discomfort (in that order) for each condition immediately after each of the six sets in both HL and LL-BFR protocols.

### 2.10. Toe-Tip Capillary Blood Lactate Analysis

Capillary blood samples were drawn from subject toe-tip using aseptic techniques via a commercially available lancet on the last visit prior to experimental HL/LL-BFR resistance exercise (PRE), immediately after completion of all six one-leg press sets (POST0), as well as 30 min (POST30M) and 3 h post (POST3H) exercise cessation. For the immediate post sampling time point, the LL-BFR condition leg remained occluded until determination of blood lactate concentration. Furthermore, previous data has demonstrated the toe as a suitable capillary blood sampling site amongst the more commonly used fingertip and earlobe (i.e., no significant differences between sampling sites during whole-body exercise) [48]. A Lactate Plus (L^+^, Nova Biomedical, Waltham, MA, USA) hand-held portable lactate analyzer, which has been validated against a laboratory blood gas analyzer, drawing ~0.7µL via test strips that do not require calibration codes or specific calibration strips [49]. The portable unit was operated in accordance with manufacturer instructions and tested against provided quality control solutions (level 1: 1.0–1.6 mM; level 2: 4.0–5.4 mM).

### 2.11. Venipuncture

Venous blood samples were obtained in 10 mL vacutainer tubes using a 21-gauge phlebotomy needle inserted into the antecubital vein. Blood samples were allowed to stand at room temperature for 10 min and then centrifuged at 2500 rpm for 15 min. The serum was then removed and immediately frozen at −80 °C for later analysis. Three blood samples were obtained during the course of the study, specifically collected at PRE and POST30M on the last/experimental visit. The former blood sample served as the baseline measurement for both exercise conditions. Furthermore, the PRE blood draw sample was drawn into micro-hematocrit tubes by capillary action and sealed with clay material [39]. These tubes were then spun for 2 min before removing and subsequently analyzed on a hematocrit reader card. Normal ranges for adult males were considered between 42–52% packed cell volume (%PCV) [39].

### 2.12. Skeletal Muscle Biopsy Procedures and Tissue Processing for cDNA Synthesis

All participants received 1 mL of anesthetic (1% lidocaine without epinephrine) via subcutaneous administration at the mid-muscle belly (half-way between the greater trochanter and patella) of the vastus lateralis. Participants then underwent a resting biopsy at the pilot hole site using the 14-gauge fine needle aspiration method using a TRU-CORE^®^ 1 Automatic Disposable Biopsy Instrument (Angiotech, Medical Device Technologies, INC., Gainsville, FL, USA) as previously described by our laboratory [50]. The biopsy needle was inserted into the pilot hole at a depth of ~5–10 mm for 2–3 passes. As a result, muscle samples totaling ~30 mg (2–3~10–15 mg samples collected per participant) were separated from connective tissue and/or adipose tissues, immediately frozen in liquid nitrogen before storing at −80 °C for later analysis. Four muscle (two on each leg) samples were obtained at the experimental visit. Biopsies were collected at PRE and POST3H following both HL and LL-BFR conditions.

Total cellular RNA was subsequently extracted from biopsy samples with a monophasic solution of phenol and guanidine isothiocyanate contained within TRI-reagent (Sigma Chemical Co., St. Louis, MO, USA). From this, 2 μg of total skeletal muscle RNA was reverse-transcribed to synthesize cDNA using the iScript cDNA Synthesis Kit (Bio-Rad, Hercules, CA, USA). Starting cDNA template concentration was standardized by adjusting all samples to 200 ng prior to Real-Time Quantitative Polymerase Chain Reaction (PCR) amplification.

### 2.13. Serum Growth Hormone (GH), Insulin-like Growth Factor-1 (IGF-1), and Homocysteine (HCY)

Serum GH and IGF-1 were assessed via commercially available enzyme-linked immunosorbent assay (ELISA) kits (DRG International, Inc., Springfield, NJ, USA), whereas serum HCY was examined using a fluorometric assay (Sigma-Aldrich, St. Louis, MO, USA) and both subsequently analyzed with a microplate reader and associated software (Infinite Pro 200, Tecan, Austria). Sample absorbance was read at a wavelength of 450 nm for GH and IGF-1, as well as an excitation/emission wavelength of 658 nm/708 nm for HCY. Moreover, unknown concentrations determined by linear regression against known standard curves using commercial software (Infinite Pro 200 with i-control™, Tecan, Austria). The average intra-assay and inter-assay coefficients of variation (CV%) for GH were 0.35% and 1.36%, respectively. Likewise, the average intra-assay and inter-assay CV% for IGF-1 were 0.5% and 1.62%, respectively. Finally, the average intra-assay and inter-assay CV% for HCY were 5.38% and 7.01%, respectively.

### 2.14. Skeletal Muscle HIF-1A Gene Expression Assessment

Real-time PCR was performed using a LightCycler 480 instrument with predesigned Applied Biosystems TaqMan™ Gene Expression Assays (Theromofisher, Waltham, MA, USA) and a TaqMan™ Fast Advanced Master Mix (Theromofisher, Waltham, MA, USA). The real-time PCR consisted of pre-denaturation followed by 45 cycles of denaturation at 95 °C for 3 s, annealing at 60 °C for 30 s, and elongation at 72 °C for 15 s. Relative quantification of the mRNA level was performed based on the second derivative maximum method (Roche, Munich, Germany). The abundance of *HIF-1A* was then normalized to a reference gene (*GAPDH*; glyceraldehyde-3-phosphate dehydrogenase). The following intron spanning assays were used: Hs00153153_m1 for *HIF-1A* and Hs02758991_g1 for *GAPDH*. The amplicon lengths were 76 and 93 bp, respectively.

### 2.15. Statistical Analyses

Prior a priori power analysis determined that a total of 16 subjects was necessary to achieve an anticipated η_p_^2^ = 0.40, and power (1-ß) = 0.80 at α = 0.05. All variables were tested for normality and homogeneity of variance using the Shapiro–Wilks test and Levene’s test of homogeneity of variance, as well as Mauchly’s test of Sphericity (when applicable) before continuing subsequent statistical analysis. Second visit 1RM was analyzed using a 2 × 2 (leg [right, left] × supplement [BET, PLA]) mixed model factorial analysis of variance (ANOVA) with repeated measures. Relative dietary macronutrients and fiber were analyzed using separate 2 × 2 (visit [visit 2, visit 3] × supplement [BET, PLA]) mixed model factorial ANOVA with repeated measures. Likewise, total repetitions across all sets was analyzed using a 2 × 2 (supplement × condition [HL, LL-BFR]) mixed model factorial ANOVA with repeated measures. Leg-specific (right vs. left) 1RM and 1RM attempt number, AOP, as well as %PCV were analyzed using individual independent t-tests for any potential between-supplement differences. Given both subjective RPE and CR10+ discomfort scores display inherently nominal data, set-specific values were collapsed; subsequent Wilcoxon signed-rank and Mann–Whitney U nonparametric tests were employed to analyze any potential supplement-collapsed exercise condition-specific and condition-collapsed supplement-specific effects, respectively. Toe-tip lactate concentrations were assessed as PRE-to-POST30M/POST0/POST3H change scores (∆Lactate) to provide simpler and more easily interpretable outcomes and were assessed via a 2 × 2 × 3 (supplement × condition × time [PRE-POST30M, PRE-POST0, PRE-POST3H]) mixed model factorial ANOVA with repeated measures [51]. Baseline serum GH, IGF-1, and HCY were analyzed for any between-supplement differences using independent t-tests. Similar to lactate, serum GH, IGF-1, and HCY were assessed as PRE-to-POST30M changes to facilitate interpretation (∆GH, ∆IGF-1, ∆HCY), whereby a 2 × 2 (supplement × condition) mixed model ANOVA with repeated measures was employed to assess any main and/or interaction effects. Lastly, relative *HIF-1A* gene expression (described as changes in gene expression from PRE to POST3H) was analyzed via a 2 × 2 (supplement × condition) mixed model ANOVA with repeated measures.

Upon any significant ANOVA model main or interaction effects, pairwise comparison analyses were employed with a Bonferroni adjustment for alpha inflation. Partial Eta squared (η_p_^2^) was used to estimate the proportion of variance in the dependent variables explained by the independent variable. Partial Eta squared effect sizes are determined to be: weak = 0.17, medium = 0.24, strong = 0.51, very strong = 0.70 [52]. Any dependent variable failing to meet normality and/or homogeneity assumptions were assessed using the appropriate non-parametric tests. Nonparametric effect size is reported as the coefficient of determination (*R*^2^), whereby *R*^2^ = 0.01, small, *R*^2^ = 0.09, medium, *R*^2^ = 0.25, large effects [51]. All analyses were performed in SPSS V.27 (IBM Corporation; Armonk, NY, USA) a significance level of *p* < *0*.05 and values reported as means ± standard deviations (SD). Confidence intervals (CI) for significant comparisons are reported as 95% CI (lower bound, upper bound).

## 3. Results

### 3.1. Subject Descriptives, Hematocrit/Packed Cell Volume (PCV%), Body Composition, and Aterial Occlusion Pressure (AOP)

All participant descriptive data, anthropometrics, and hemodynamic variables are displayed in Table 1. Additionally, participant AOP and PCV% can both be viewed in Table 2. There were no statistically significant between-supplement differences between either of the aforementioned variables. Furthermore, all 18 participants demonstrated 100% supplement compliance.

### 3.2. Dietary Assessments

As determined by our investigative team’s RD, all but two participants were required to administer a daily b-vitamin complex in addition to their respective supplement condition. Dietary data for both visit 2 (1RM assessment) and visit 3 (final visit/experimental testing) are displayed in Table 3. Briefly, analyses demonstrated no statistically significant differences between visits, supplement conditions, nor any visit-supplement interaction effects.

### 3.3. 1RM Determination and Repetitions to Failure

Participant right and left leg 1RM, as well as associated 1RM attempt numbers are displayed in Table 2. Analyses revealed no significant between-supplement group differences for any of the aforementioned variables. Incidentally, a total of 4 (22.2%) and 3 (16.7%) participants were unable to achieve the standard repetitions prescribed for the HL (40 total) and LL-BFR (75 total) condition, respectively. The lowest number of repetitions accomplished amongst these participants (not including the repetitions to failure on sets 5 and 6) were 26 and 51 for HL and LL-BFR, respectively. Nevertheless, all other participants were able to complete the full prescribed set and repetition protocol. Analysis failed to reveal any significant supplement main (*p* = 0.866; η_p_^2^ = 0.002) or exercise × supplement interaction (*p* = 0.398; η_p_^2^ = 0.048) effects, although a significant “very strong” main exercise condition (*p* < 0.001; η_p_^2^ = 0.740) effect was detected. Specifically, LL-BFR performed significantly more total repetitions relative to HL (*p* < 0.001; CI [26.273, 53.764]). The total repetitions variable nevertheless failed normality assumptions, whereby a nonparametric Wilcoxon signed-rank test confirmed a large aforementioned significant effect (*p* < 0.001; *R*^2^ = 0.376).

### 3.4. RPE and Subjective Discomfort

Nonparametric analyses failed to detect a significant supplement-collapsed exercise condition (*p* = 0.743) nor condition-collapsed supplement (*p* = 0.351) effects for RPE. Conversely, nonparametric analyses revealed a significant medium-large supplement-collapsed exercise condition effect for CR10+ subjective discomfort (*p* < 0.001, *R*^2^ = 0.201), but did not demonstrate a significant condition-collapsed supplement effect (*p* = 0.977). Combined-set LL-BFR specifically displayed a greater average subjective discomfort relative to HL (mean CR10+ score 6.5 ± 2.8 [LL-BFR] versus 5.2 ± 2.6 [HL]). The overall changes in subjective RPE and CR10+ scores can be visualized in Figure 2A,B.

### 3.5. 1RM Toe-Tip Capillary Lactate

Three-way analyses detected significant exercise condition (*p* = 0.003; η_p_^2^ = 0.432) and time (*p* < 0.001; η_p_^2^ = 0.810) main effects for ∆Lactate. Conversely, there were no significant supplement effect (*p* = 0.707; η_p_^2^ = 0.009) nor any interaction effects (condition × supplement [*p* = 0.555; η_p_^2^ = 0.022], condition × time [*p* = 0.284; η_p_^2^ = 0.074] time × supplement [*p* = 0.684; η_p_^2^ = 0.023], condition × time × supplement [*p* = 0.996; η_p_^2^ = 0.002]). See Table 4 for a detailed description of all raw and ∆Lactate data, as well as all pairwise comparisons.

### 3.6. Serum GH, IGF-1, and HCY Assessment

All serum GH, IGF-1, and HCY data is displayed in Table 5. Baseline serum IGF-1 and HCY were not different between supplement conditions (*p* = 0.117). Incidentally, baseline serum GH failed normality assumptions but a nonparametric Mann–Whitney U test determined no significant supplement-specific baseline differences (*p* = 0.566). Analyses failed to reveal any significant exercise condition (*p* = 0.256; η_p_^2^ = 0.091), supplement (*p* = 0.207; η_p_^2^ = 0.111), or interaction (*p* = 0.487; η_p_^2^ = 0.035) effects for serum ∆GH. Similarly, there were no exercise condition (*p* = 0.452; η_p_^2^ = 0.038) or interaction effects (*p* = 0.666; η_p_^2^ = 0.013), however, analyses demonstrated a significant medium supplement effect for ∆IGF-1 (*p* = 0.042; η_p_^2^ = 0.247), whereby BET displayed significantly higher serum IGF-1 changes from baseline relative to PLA (*p* = 0.042; CI [0.752, 36.657 ng·mL^−1^]). Analyses further failed to reveal any significant main supplement (*p* = 0.184; η_p_^2^ = 0.107) nor interaction (*p* = 0.162; η_p_^2^ = 0.119) effects for ∆HCY. Conversely, there was a statistically significant main exercise condition (*p* = 0.045; η_p_^2^ = 0.228), whereby the supplement-collapsed HL-versus-LLO group demonstrated greater changes in HCY concentrations at POST30 (*p* = 0.045; CI [0.089, 7.300 μmol·mL^−1^]). Serum ∆IGF-1 and ∆HCY nonetheless failed normality assumptions, and thus the aforementioned significant findings were confirmed via non-parametric Mann–Whitney U and Wilcoxon signed rank tests (∆IGF-1: *p* = 0.029, *R*^2^
*=* 0.132; ∆HCY: *p* = 0.044; *R*^2^
*=* 0.113), respectively.

### 3.7. HIF-1A Gene Expression Assessment

*HIF-1A* transcript abundance is displayed in Table 5. Moreover, the change in *HIF-1A* gene expression from PRE to POST3H was not statistically significant between exercise conditions (*p* = 0.287; η_p_^2^ = 0.087), supplement groups (*p* = 0.635; η_p_^2^ = 0.018), nor were any significant interaction effects (*p* = 0.830; η_p_^2^ = 0.004) present.

## 4. Discussion

The present investigation aimed to assess the viability of a synergistic BFR-betaine supplementation combination amongst several resistance training-associated parameters. Contrary to our hypothesis, the interaction between these distinct modalities did not cumulatively provide any additive benefit across the assessed outcomes. Although analyses were unable to detect any synergistic or supplement-specific differences, LL-BFR demonstrated a higher number of total repetitions relative to HL as expected. Interestingly, this was not accompanied by a greater BFR-associated ∆Lactate. The present lactate data largely trended as expected, reaching peak levels before subsequently dropping towards baseline concentrations at POST3H. It is also worthwhile to note the mean baseline lactate concentrations in all groups were above typical resting levels, but this phenomenon maybe due to protocol-associated psychosocial stress incurred on our participants [53]. Nonetheless, the HL group demonstrated a significantly supplement-collapsed elevation in lactate concentrations relative to LL-BFR. Provided capillary lactate assessment is positively associated with exercise intensity-mediated RPE scaling, we assumed our subjective measures would support this relationship; however, exertion did not significantly differ between any supplement or exercise condition combination and only HL displayed higher subjective discomfort [54]. Notwithstanding these conflicting results, the present findings do match those of Kim et al. [11], whereby the authors discovered greater lactate concentrations from baseline to immediately-post leg press and knee extension exercise in a high-load (80%1RM) versus low-load (20%1RM) BFR condition, ultimately contending that the difference was likely due to the higher total loads of the former. It is additionally possible that HL exercise incurred a greater intensity-dependent catecholamine response, whereby previous investigations such as Takano et al. [15] displayed BFR-specific increases relative only to an equivalently loaded control (20%). Increased epinephrine has previously been demonstrated to correlate highly with commensurate serum lactate via cyclic adenosine monophosphate-mediated phosphorylase a activation and successive glycogenolysis [55,56,57,58]. This phenomenon may therefore have facilitated the higher HL-versus-LL-BFR condition-associated lactate concentrations observed in the current investigation. Consequently, the supplement-collapsed HL-specific serum HCY concentrations may be explained by this phenomenon as well, considering norepinephrine-to-epinephrine biosynthesis facilitates methyl metabolism flux via enzymatic phenylethanolamine methyltransferase action [59]. Future investigations would nevertheless benefit from assessing serum epinephrine, as well as muscle glycogen concentrations to elucidate differences in substrate utilization and concomitant lactate changes. It is also pertinent to note that the greater HL-specific group-level lactate concentrations were not accompanied by ostensibly associated GH and IGF-1 elevations. Regardless, the data reporting on this relationship is—as previously stated—mixed; [12,13,14,15,16,60]. Incidentally, the methodology employed amongst these associated investigations is equally diverse, whereby many of the prior studies displaying a superior BFR-associated lactate singularly compare to uncuffed low-load conditions [13,15,61,62]. Similar to the present findings, trials examining the impact of BFR against high-load modalities ultimately demonstrate varied results [11,12]. These disparate outcomes warrant additional research to elucidate the extent to which BFR training elicits increases in GH and IGF-1 when considering its unique metabolic and load-related attributes.

In opposition to our original hypotheses, *HIF-1A* gene expression was not differentially impacted by a combined BFR-betaine modality. Although it appears that betaine did not facilitate a greater BFR exercise-mediated hypoxic stimulus, the present findings build upon mixed prior evidence. Drummond et al. [63] specifically displayed similar *HIF-1A* expression between equivalently loaded (20%1RM) BFR and non-inflated leg exercise, crediting these equivocal findings to their protocol’s supposedly more metabolic versus hypoxic stimulus. Conversely, Laurentino et al. [64] observed an inferior *HIF-1A* expression in uncuffed low-load versus likewise loaded BFR and high-load bilateral knee extensions, with no difference between the latter conditions. This potential load-associated stimuli discrepancy remains unclear, whereby the repetitions to muscular failure employed within the current investigation may have created a superseding metabolically associated stimulus that impaired our ability to detect any between- supplement or- exercise condition main, nor any possible synergistic interaction effects [63,65]. Incidentally, more recent data indicate HIF-1 signaling is significantly impaired following initial training adaptations via enhanced HIF-1 repressors (i.e., several prolyl hydroxylase isoenzymes, factor inhibiting HIF-1 [FIH], and sirtuin 6) [66]. We accordingly posit that our equivocal findings may then have been from recruiting already trained participants with similar experience over the last year at minimum. Moreover, and despite the apparent non-significant enhancements in *HIF-1A* gene expression, it is nevertheless possible that we constrained our investigation to a single target. Although the former was chosen as a more encompassing metabolic adaptive mediator, it would be sensible to further assess other relevant genes such as those involved in fiber type alteration and/or lactate-specific metabolism that already have mechanistic rationale within the BFR-associated literature [2,3,8,67,68]. The combination of BFR exercise and betaine supplementation may overall have untried applications amidst a more broad genetic target range, possibly further elucidating the divergent role of HIF-1 within BFR-associated adaptations.

## 5. Conclusions

### Limitations and Final Remarks

While the present study sought to explore the viability of a combined BFR-betaine synergy, it was limited by several factors. The acute nature of our study design does not permit the current data to be extrapolated to longitudinal outcomes. While both serum GH and IGF-1 increased following one-leg press resistance exercise, only the latter demonstrated a betaine supplementation-specific advantage without any discernable BFR-associated impact. Incidentally, homocysteine is negatively correlated to IGF-1, ostensibly impairing the GH-IGF-1 axis [69,70,71]. Furthermore, GH hypothetically augments IGF-1 synthesis via priming associated open chromatin configurations [72]. Betaine may then consequently augment the GH-IGF-1 axis by facilitating comprehensive metabolic and epigenetic methylation processes [72,73]. These findings therefore largely support a betaine supplementation-enhanced anabolic signaling capacity, whereby IGF-1 is known to bind to the insulin receptor and facilitate downstream translational efficiency via the phosphatidylinositol-3 kinase (PI3K)/Akt/mammalian target of rapamycin (mTOR) complex pathway [74]. Apicella et al. [17] demonstrated similar findings to the present study, illustrating equivocal serum GH and a (2.5 g/day) betaine-specific IGF-1 increase. The latter increase was commensurate with a parallel enhancement in ribosomal protein S6 beta-1 kinase (p70S6K), a downstream PI3K/Akt/mTOR target [17]. Although we are unable to claim these findings taken cumulatively would infer enhanced longitudinal hypertrophy, previous literature has otherwise substantiated this notion [18,20]. Nevertheless, future research should aim to assess betaine-mediated IGF-1 increases and how they impact long-term whole-muscle growth across several training sessions.

Previous authors have individually stipulated that BFR exercise and betaine supplementation likely exert their positive isolated adaptations particularly through long-term training [20]. Specifically, employing these training aids may require at least six days to four weeks for hypertrophic adaptations to discernably manifest in BFR exercise and betaine supplementation, respectively [13,20,75]. Although our subjects supplemented with betaine for 2-weeks, the actual combined BFR-betaine modality was only acutely implemented on a single day. Our analysis was therefore not able to assess additional factors that would be reasonably impacted over a longer timeframe. Given the multiplicity of metabolically relevant proteins and preceding genes that could be impacted distinctly by either modality, the longitudinal execution of a BFR-betaine investigation is warranted [2]. While the present study failed to detect a significant synergistic advantage, these data substantiate a potential betaine-specific anabolic environment. Nevertheless, our findings only begin to explore this combination. Future research is hereupon tasked to build upon the current work, ultimately exposing the full breadth of plausibly augmented metabolic adaptations amidst an equally exhaustive timeframe.

## Figures and Tables

**Figure 1 nutrients-14-05040-f001:**
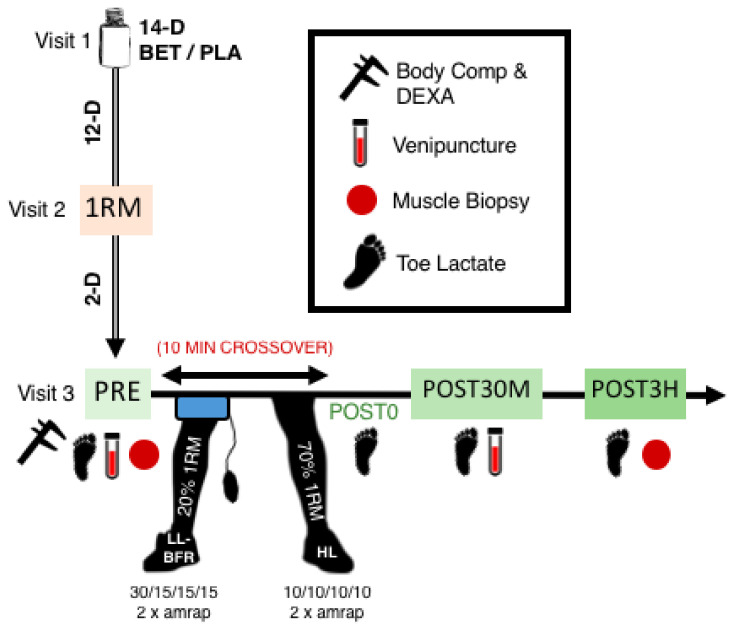
Study protocol and timeline visualization. 1RM = one-repetition maximum; amrap = as many repetitions as possible (to muscular failure); BET = betaine; DEXA = dual-energy x-ray absorptiometry; HL = high-load; LL-BFR = low-load blood flow restricted; PLA = placebo.

**Figure 2 nutrients-14-05040-f002:**
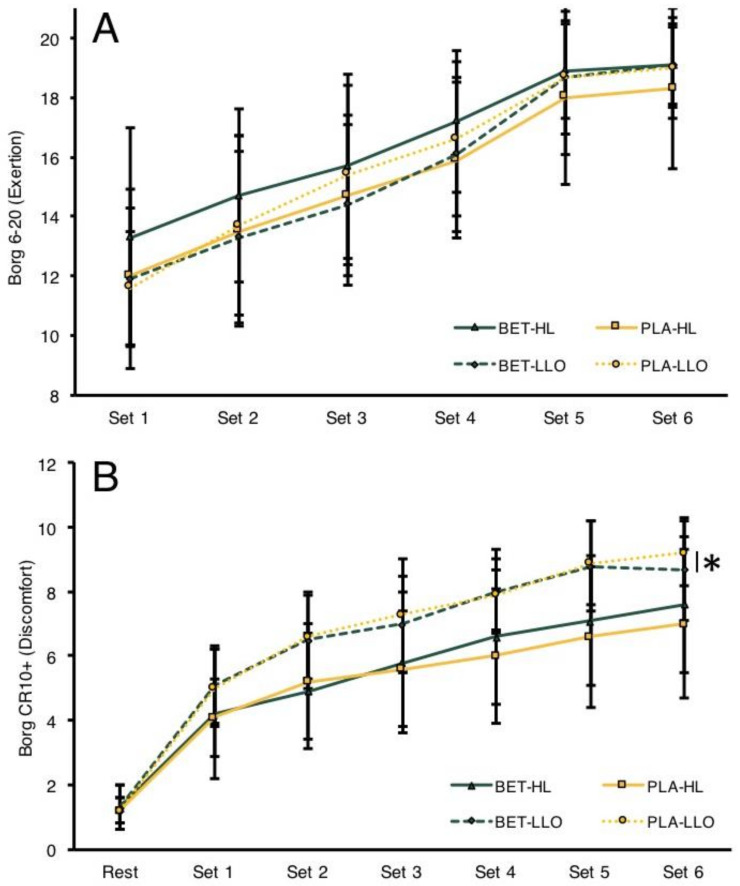
Rated perceived exertion (RPE; (**A**) changes from sets 1 through 6 and subjective discomfort (CR10+; (**B**) changes from resting through set 6. All data are displayed as means ± SD. * Indicates significant supplement-collapsed exercise condition effect (*p* < 0.001), whereby LL-BFR displayed greater subjective discomfort versus HL.

**Table 1 nutrients-14-05040-t001:** Participant demographics, anthropometrics, and resting hemodynamic parameters.

Mean ± SD	BET (n = 9)	PLA (n = 9)
Age (years)	23 ± 3	22 ± 3
Height (cm)	177.9 ± 6.5	179.5 ± 7.7
Weight (kg)	87.6 ± 5.4	84.2 ± 17.3
BF%	15.6 ± 3.0	13.5 ± 3.6
Resting HR (bpm)	68 ± 8	68 ± 7
Resting SBP/DBP (mmHg)	125 ± 10/74 ± 13	128 ± 9/69 ± 9

BET = betaine; BF = body fat percentage; HR = heart rate; PLA = placebo.

**Table 2 nutrients-14-05040-t002:** Second (1RM assessment) and third (experimental protocol/final) visit hematocrit/packed cell volume % (PCV%) assessment, LL-BFR condition AOP, HL/LL-BFR prescribed load data. All data are presented as means ± SD. In brief, there were no significant between-leg differences for 1RM nor 1RM attempts during visit 2. Likewise, neither AOP nor PCV% differed between supplement conditions.

Mean ± SD	BET (n = 9)	PLA (n = 9)	*p*-Value (<0.05)
1RM Testing (Supplementation Day 12)			
1RM RL (kg)	263 ± 46	251 ± 72	Main Leg Effect = 0.802 Interaction Effect = 0.819
1RM LL (kg)	260 ± 53	251 ± 81	
Total 1RM Attempts RL	5 ± 1	6 ± 2	0.200
Total 1RM Attempts LL	5 ± 2	6 ± 2	0.200
Experimental Protocol (12 h Post Supplementation)			
AOP (mmHg)	318 ± 82	310 ± 62	0.833
PCV%	45.9 ± 1.6	46.9 ± 2.9	0.384

1RM = one-repetition maximum; AOP = arterial occlusion pressure; BET = betaine; LL = left leg; PCV% = % packed cell volume; PLA = placebo; RL = right leg.

**Table 3 nutrients-14-05040-t003:** Participant dietary macronutrient and fiber intake reported as average relative consumption (g·kg^−1^ bodyweight) for the 24 and 48 h preceding both the second (1RM assessment) and third (experimental protocol) visit, respectively. All data are presented as means ± SD. In brief, there were no significant between-visit, supplement, nor interaction effects observed.

Mean ± SD	BET (n = 9)	PLA (n = 9)	*p*-Value (<0.05); ES (η_p_^2^)
Dietary CHO (g·kg^−1^ bw)			Main Supplement Effect = 0.388; η_p_^2^ = 0.042
1RM Testing	3.0 ± 0.9	3.2 ± 1.0	Main Time Effect = 0.416; η_p_^2^ = 0.042
Experimental Testing	2.7 ± 1.0	3.2 ± 1.3	Interaction Effect = 0.451; η_p_^2^ = 0.036
Dietary PRO (g·kg^−1^ bw)			Main Supplement Effect = 0.502; η_p_^2^ = 0.936
1RM Testing	1.6 ± 0.6	1.7 ± 0.4	Main Time Effect = 0.334; η_p_^2^ = 0.058
Experimental Testing	1.6 ± 0.6	1.8 ± 0.5	Interaction Effect = 0.863; η_p_^2^ = 0.002
Dietary FAT (g·kg^−1^ bw)			Main Supplement Effect = 0.078; η_p_^2^ = 0.181
1RM Testing	0.9 ± 0.2	1.3 ± 0.4	Main Time Effect = 0.666; η_p_^2^ = 0.012
Experimental Testing	1.0 ± 0.4	1.3 ± 0.8	Interaction Effect = 0.666; η_p_^2^ = 0.012
Dietary Fiber (g·kg^−1^ bw)			Main Supplement Effect = 1.00; η_p_^2^ = 0.835
1RM Testing	0.3 ± 0.1	0.2 ± 0.1	Main Time Effect = 0.429; η_p_^2^ = 0.040
Experimental Testing	0.2 ± 0.1	0.2 ± 0.1	Interaction Effect = 0.124; η_p_^2^ = 0.142

1RM = one-repetition maximum; BET = betaine; bw = bodyweight; CHO = carbohydrate; ES = effect size; PLA = placebo; PRO = protein.

**Table 4 nutrients-14-05040-t004:** Raw lactate and ∆Lactate (change from rest/baseline) concentrations (nmol/L). All data are presented as means ± SD.

Mean ± SD Lactate (mmol·L^−1^)	BET				PLA			
	PRE	POST0	POST30M	POST3H	PRE	POST0	POST30M	POST3H
HL	2.7 ± 0.7	6.9 ± 2.3	6.4 ± 1.9	2.9 ± 1.0	2.4 ± 0.7	6.1 ± 1.2	6.2 ± 1.7	2.7 ± 1.6
LL-BFR	2.3 ± 0.8	6.1 ± 1.2	5.0 ± 1.6	2.7 ± 1.1	2.5 ± 0.8	5.9 ± 1.6	5.3 ± 1.6	2.7 ± 1.1
		∆POST0	∆POST30M	∆POST3H		∆POST0	∆POST30M	∆POST3H
HL		4.1 ± 2.2 *	3.7 ± 2.1 *	0.2 ± 0.9 *^,^†		3.7 ± 1.3 *	3.8 ± 1.7 *	0.3 ± 1.1 †
LL-BFR		3.8 ± 2.6	2.6 ± 1.7	0.6 ± 0.5 †		3.4 ± 1.4	2.8 ± 1.4	0.2 ± 0.8 †

BET = betaine; HL = high-load; LL-BFR = low-load blood flow restricted PLA = placebo; PRE = before resistance exercise; POST0 = immediately post-exercise; POST30M = 30 min post-exercise; POST3H = 3 h post-exercise. *: HL displayed significantly higher condition- and time-collapsed ∆Lactate (*p* = 0.003, CI [0.160, 0.655 mmol·L^−1^]) compared to LL-BFR. †: The change in lactate from baseline concentrations was significantly lower at POST3H relative to both POST0 (*p* < 0.001, CI [−4.324, −2.588 mmol·L^−1^]) and POST30M (*p* < 0.001, CI [−3.800, −2.033 mmol·L^−1^]), whereby the latter two did not significantly differ from one another (*p* = 0.272).

**Table 5 nutrients-14-05040-t005:** Serum GH, IGF-1, and HCY concentrations, as well *HIF-1A* transcript levels across all condition and supplement combinations. All data are presented as means ± SD. * Condition-collapsed BET displayed significantly higher IGF-1 concentrations relative to PLA and † Supplement-collapsed HL-versus-LLO demonstrated significantly greater HCY concentrations (*p* < 0.05).

Mean ± SD	BET		PLA		Supplement-Collapsed	
	PRE	POST30M	PRE	POST30M		
GH (ng·mL−1)						
*HL*		3.64 ± 3.01		5.51 ± 5.23	PRE	1.36 ± 2.13
*LL-BFR*		4.32 ± 2.94		7.16 ± 12.01	HL-POST30M	4.52 ± 4.17
*Condition-Collapsed*	1.76 ± 2.62	3.98 ± 2.91	0.96 ± 1.53	4.40 ± 7.63	LL-BFR-POST30M	5.66 ± 8.34
IGF-1 (ng·mL−1)						
*HL*		111.1 ± 35.2		122.7 ± 34.2	PRE	120.0 ± 42.0
*LL-BFR*		118.1 ± 31.5		134.7 ± 41.8	HL-POST30M	116.5 ± 34.2
*Condition-Collapsed*	105.4 ± 34.0	114.6 ± 32.6 *	136.6 ± 45.1	129.1 ± 37.7	LL-BFR-POST30M	126.4 ± 36.9
HCY (μmol·L−1)						
*HL*		29.1 ± 4.92		30.4 ± 3.85	PRE	28.2 ± 2.96
*LL-BFR*		27.9 ± 4.17		27.2 ± 3.12	HL-POST30M	29.7 ± 4.34 †
*Condition-Collapsed*	27.7 ± 2.50	28.5 ± 4.47	28.8 ± 3.42	28.9 ± 3.79	LL-BFR-POST30M	27.6 ± 3.61
*HIF-1A transcript levels*						
*HL*	0.42 ± 0.28	0.62 ± 0.49	1.40 ± 2.67	3.76 ± 9.21	HL-PRE	0.86 ± 1.90
*LL-BFR*	0.45 ± 0.23	0.45 ± 0.23	1.48 ± 3.40	0.73 ± 0.68	LLO-PRE	0.96 ± 2.40
*Condition-Collapsed*	0.43 ± 0.25	0.76 ± 0.58	1.44 ± 2.98	2.24 ± 6.52	HL-POST30M	2.19 ± 6.53
					LL-BFR-POST30M	0.82 ± 0.70

BET = betaine; GH = growth hormone; HCY = homocysteine; HIF-1A = hypoxia-inducible factor 1α; HL = high-load; IGF-1 = insulin-like growth factor-1; LL-BFR = low-load blood flow restricted; PLA = placebo; PRE = before resistance exercise; POST30M = 30 min post-exercise.

## Data Availability

The data presented in this study are available on request from the corresponding author.

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
