# Peer review of "The Impacts of Combined Blood Flow Restriction Training and Betaine Supplementation on One-Leg Press Muscular Endurance, Exercise-Associated Lactate Concentrations, Serum Metabolic Biomarkers, and Hypoxia-Inducible Factor-1α Gene Expression"

_nutrients, 2022, doi:10.3390/nu14235040_

Round 1

Reviewer 1 Report

The authors seek to describe the experimental work indicated in the title of the thesis.

It can be concluded that the “Effect of Combined Blood Flow Restriction Training an Betain on One-Leg Press Performance” part indicated in the title is not accurately presented. The initial performance, the effect of intervention, is not known in the article. The additional information on the measurement is correct, but it also provides a number of information that will not be discussed later. Less would be more.

To the extent that the authors insist on the original title, I see a need to make the following change:

1. Was there an RFL training and what did it consist of?

2. How was One-Leg Press Performance actually measured, with what device? What parameters were recorded?

3. PRE values should be treated with caution, lactate values draw attention to this. PRE values are not resting values.

4. After the body composition has been made using the DEXA method, with the help of a suitable program, it will be possible to determine the muscle mass of the tested part of the body (ROI). This may play an important role in the supplementation and explanation of other load-induced biomarker changes.

5. I suggest omitting the information presented that is not discussed in the thesis.

6. It would be very useful, If the authors give some information about he assumed effect of the RFL on the anaerobic metabolism and adaptation characteristics from special point of view of the measured and discussed biomarkers.

6. In the conclusion, many theoretic findings and propositions are made, a reduced presentation of the results would suffice.

Author Response

Dear Reviewer #1,

We would like to thank you for your critical feedback and valuable suggestions. Whenever possible, we have made edits to match your recommendations and have attempted to substantiate our methods where we may have been unclear. We kindly implore you to revisit the updated iteration of our manuscript and respond with any additional suggestions or inquires you might have.

The authors seek to describe the experimental work indicated in the title of the thesis.

It can be concluded that the “Effect of Combined Blood Flow Restriction Training an Betain on One-Leg Press Performance” part indicated in the title is not accurately presented. The initial performance, the effect of intervention, is not known in the article. The additional information on the measurement is correct, but it also provides a number of information that will not be discussed later. Less would be more.

With respect to the title, we understand that the term “performance” might have been an overstatement. We have thus changed it to “…One-Leg Press Muscular Endurance” to better represent the purpose and hypotheses in our introduction.

To the extent that the authors insist on the original title, I see a need to make the following change:

  1. Was there an RFL training and what did it consist of?

Response: Under the assumption that RFL refers to “Required Fitness Level”, the authors would like to direct the reviewer to section 2.2, Page 4, Lines 170-172, whereby we state that the participants were required to exercise loosely based on the current ACSM guidelines: “<30 minutes exercise, <3 days per week, over the last 3 months, as well as having a minimum of one day per week leg-focused exercise over the last year prior to the onset of the investigation”. Additionally, we added a segment to section 2.6, Page 5, Lines 232-234 that details a 1x bodyweight one-leg press strength minimum. This metric was based on an adaptation by a dissertation-based investigation by Ramsey et al. (2018), which was applied to the methodology strength standards methodology described by Hoffman (2017). Please let us know if you had another intention for the term ,”RFL” and we will absolutely make edits accordingly.

  1. How was One-Leg Press Performance actually measured, with what device? What parameters were recorded?

Response: The specific leg press manufacturer is now included on Page 5, Line 231. With regards to performance parameters, sections 2.6 and 2.8 describe the specific metrics assessed. The former denotes the methods by which we collected right and left leg one-leg press maximal 1RM strength. Conversely, the latter details that the total number of repetitions performed for both HL and LL-BFR conditions were recorded and subsequently compared for any potential between exercise-condition differences.

  1. PRE values should be treated with caution, lactate values draw attention to this. PRE values are not resting values.

Response: We acknowledge that the mean resting lactate values are indeed slightly over the normal range >2mmol/L. We accredit this phenomenon to psychosocial factors that have been previously noted to elevate resting values as described by Kubera et al. 2012 (PMID: 22797365). A sentence had been added to the discussion section detailing this potential discrepancy. We would additionally like to affirm that our values are accurate, considering the capillary Lactate assessment tool we employed was tested against quality control solutions prior to each subject visit. Moreover, the present investigation was more so focused on the changes in capillary lactate concentrations to compare supplement and exercise conditions.

  1. After the body composition has been made using the DEXA method, with the help of a suitable program, it will be possible to determine the muscle mass of the tested part of the body (ROI). This may play an important role in the supplementation and explanation of other load-induced biomarker changes.

Response: We recognize the value of looking specifically at the body part of interest as you suggest. Unfortunately, the investigators are no longer able to access the data at the location of data collection for further analysis. We would moreover contend that although beneficial, DEXA-specific body composition was largely assessed to determine any major body composition differences between supplement groups that may have impacted outcomes such as one leg press strength and/or repetition number, as well as other markers associated with inflammation (i.e. homocysteine). There would be particular benefit to your suggestion if the present investigation was a longitudinal design, but its otherwise cross-sectional nature likely does not offer any further applicable information.

  1. I suggest omitting the information presented that is not discussed in the thesis.

Response: We apologize in that we are uncertain which information you are specifically referring to that is suggested for omission, but as per your recommendation in #7, we have made several adjustments to our manuscript to create a more abbreviated narrative. We hope that is addresses your concern and once again invite further suggestions to assist in improving the current iteration of our work.

  1. It would be very useful, If the authors give some information about he assumed effect of the RFL on the anaerobic metabolism and adaptation characteristics from special point of view of the measured and discussed biomarkers.

Response: Although useful, the authors of the current manuscript would contend that additional information on how the required fitness level of the participants impacts these markers would be outside of the scope of our presentation. Notwithstanding the need to describe the comparative effects of both a supplement and distinct exercise modalities, we believe that a description of how training history potentially impacted our present outcome measures would ultimately dilute our interpretations by unnecessarily extending our introduction and/or discussion sections. We believe the information you have suggested to include would be especially relevant if comparing untrained to trained participants, however, our subjects share a relatively homogenous training experience history as per our inclusion criteria.

For example, aspects such as the presently observed HL-specific serum homocysteine concentrations might have been due to a lower relative resistance training experience as supported by an investigation by Deminice et al. (2016; PMID 26986570). Nevertheless, no other measured parameter would have suggested any potential between-group baseline difference that implicate unequal training experiences. We instead provided a mechanistic explanation for this phenomenon relating to the greater HL-specific lactate concentrations and how both aforementioned statistically significant findings can be at least partially explained by intensity-dependent catecholamine levels. Further speculation is provided towards the bottom of page 15, whereby we speculate that our equivocal HIF-1A gene expression data might have been due to training adaptation-mediated increases in HIF suppressors.

Despite the aforementioned reasons, if you would comment on a particular outcome measure of interest that warrants RFL-specific description, we would gladly include such information.

  1. In the conclusion, many theoretic findings and propositions are made, a reduced presentation of the results would suffice.

Response: We greatly appreciate your constructive suggestions and have accordingly trimmed the discussion down. Many of the theoretic propositions made were to hopefully provide a sufficient explanation for the results of the present investigation and to provide a base for future research. Thus, we respectfully decided to maintain the general ideas and the associated supporting evidence. Many of the suppositions we have made in the discussion are purely pertinent to the investigated parameters, so we sincerely hope that you can understand that the detail we’ve included attempts to avoid significant deviations from the relevant outcomes observed.

Reviewer 2 Report

In the abstract the acronym BFR apears for the first time, and as a first time it should be as a full word 

on line 76 a "t" is needed. "...to translocate across he nuclear..." (?)

Author Response

Dear Reviewer #2

The authors of the present investigation are extremely grateful for your critical eye and the mistakes you have caught. We have acted on both of your suggestions and hope that the current iteration is now more suitable for future readership. We implore you to let us know if you catch any further errors and know that we would be happy to address them.

In the abstract the acronym BFR appears for the first time, and as a first time it should be as a full word 

Response: The full term has now been included before the acronym

on line 76 a "t" is needed. "...to translocate across he nuclear..." (?)

Response: The above mistake has now been amended

Round 2

Reviewer 1 Report

The answers are accepted, with the changes of some paragraphs, it is more easy to understand the text.